# Learning Mixtures of Submodular Functions for Image Collection Summarization

**Sebastian Tschiatschek**
Department of Electrical Engineering
Graz University of Technology
tschiatschek@tugraz.at

**Rishabh Iyer**
Department of Electrical Engineering
University of Washington
rkiyer@u.washington.edu

**Haochen Wei**
LinkedIn & Department of Electrical Engineering
University of Washington
weihch90@gmail.com

**Jeff Bilmes**
Department of Electrical Engineering
University of Washington
bilmes@u.washington.edu

## Abstract

We address the problem of image collection summarization by learning mixtures of submodular functions. Submodularity is useful for this problem since it naturally represents characteristics such as fidelity and diversity, desirable for any summary. Several previously proposed image summarization scoring methodologies, in fact, instinctively arrived at submodularity. We provide classes of submodular component functions (including some which are instantiated via a deep neural network) over which mixtures may be learnt. We formulate the learning of such mixtures as a supervised problem via large-margin structured prediction. As a loss function, and for automatic summary scoring, we introduce a novel summary evaluation method called V-ROUGE, and test both submodular and non-submodular optimization (using the submodular-supermodular procedure) to learn a mixture of submodular functions. Interestingly, using non-submodular optimization to learn submodular functions provides the best results. We also provide a new data set consisting of 14 real-world image collections along with many human-generated ground truth summaries collected using Amazon Mechanical Turk. We compare our method with previous work on this problem and show that our learning approach outperforms all competitors on this new data set. This paper provides, to our knowledge, the first systematic approach for quantifying the problem of image collection summarization, along with a new data set of image collections and human summaries.

## 1  Introduction

The number of photographs being uploaded online is growing at an unprecedented rate. A recent estimate is that 500 million images are uploaded to the internet every day (just considering Flickr, Facebook, Instagram and Snapchat), a figure which is expected to double every year [22]. Organizing this vast amount of data is becoming an increasingly important problem. Moreover, the majority of this data is in the form of personal image collections, and a natural problem is to summarize such vast collections. For example, one may have a collection of images taken on a holiday trip, and want to summarize and arrange this collection to send to a friend or family member or to post on Facebook. Here the problem is to identify a subset of the images which concisely represents all the diversity from the holiday trip. Another example is scene summarization [28], where one wants to concisely represent a scene, like the *Vatican* or the *Colosseum*. This is relevant for creating a visual summary of a particular interest point, where we want to identify a representative set of views. Another application that is gaining importance is summarizing video collections [26, 13] in order to enable efficient navigation of videos. This is particularly important in security applications, where one wishes to quickly identify representative and salient images in massive amounts of video.

These problems are closely related and can be unified via the problem of finding the most representative subset of images from an entire image collection. We argue that many formulations of this problem are naturally instances of submodular maximization, a statement supported by the fact that a number of scoring functions previously investigated for image summarization are (apparently unintentionally) submodular [30, 28, 5, 29, 8].

A set function $f(\cdot)$ is said to be submodular if for any element $v$ and sets $A \subseteq B \subseteq V \setminus \{v\}$, where $V$ represents the ground set of elements, $f(A \cup \{v\}) - f(A) \geq f(B \cup \{v\}) - f(B)$. This is called the diminishing returns property and states, informally, that adding an element to a smaller set increases the function value more than adding that element to a larger set. Submodular functions naturally model notions of coverage and diversity in applications, and therefore, a number of machine learning problems can be modeled as forms of submodular optimization [11, 20, 18]. In this paper, we investigate structured prediction methods for learning weighted mixtures of submodular functions for image collection summarization.

**Related Work:** Previous work on image summarization can broadly be categorized into (a) clustering-based approaches, and (b) approaches which directly optimize certain scoring functions. The clustering papers include [12, 8, 16]. For example, [12] proposes a hierarchical clustering-based summarization approach, while [8] uses $k$-medoids-based clustering to generate summaries. Similarly [16] proposes top-down based clustering. A number of other methods attempt to directly optimize certain scoring functions. For example, [28] focuses on scene summarization and poses an objective capturing important summarization metrics such as likelihood, coverage, and orthogonality. While they do not explicitly mention this, their objective function is in fact a submodular function. Furthermore, they propose a greedy algorithm to optimize their objective. A similar approach was proposed by [30, 29], where a set cover function (which incidentally also is submodular) is used to model coverage, and a minimum disparity formulation is used to model diversity. Interestingly, they optimize their objective using the same greedy algorithm. Similarly, [15] models the problem of diverse image retrieval via determinantal point processes (DPPs). DPPs are closely related to submodularity, and in fact, the MAP inference problem is an instance of submodular maximization. Another approach for image summarization was posed by [5], where they define an objective function using a graph-cut function, and attempt to solve it using a semidefinite relaxation. They unintentionally use an objective that is submodular (and approximately monotone [18]) that can be optimized using the greedy algorithm.

**Our Contributions:** We introduce a family of submodular function components for image collection summarization over which a convex mixture can be placed, and we propose a large margin formulation for learning the mixture. We introduce a novel data set of fourteen personal image collections, along with ground truth human summaries collected via Amazon mechanical Turk, and then subsequently cleaned via methods described below. Moreover, in order to automatically evaluate the quality of novel summaries, we introduce a recall-based evaluation metric, which we call V-ROUGE, to compare automatically generated summaries to the human ones. We are inspired by ROUGE [17], a well-known evaluation criterion for evaluating summaries in the document summarization community, but we are unaware of any similar efforts in the computer vision community for image summarization. We show evidence that V-ROUGE correlates well with human evaluation. Finally, we extensively validate our approach on these data sets, and show that it outperforms previously explored methods developed for similar problems. The resulting learnt objective, moreover, matches human summarization performance on test data.

## 2 Image Collection Summarization

Summarization is a task that most humans perform intuitively. Broadly speaking, summarization is the task of extracting information from a source that is both minimal and most important. The precise meaning of *most important* (relevance) is typically subjective and thus will differ from individual to individual and hence is difficult to precisely quantify. Nevertheless, we can identify two general properties that characterize good image collection summarizes [19, 28]:

**Fidelity:** A summary should have good coverage, meaning that all of the distinct "concepts" in the collection have at least one representative in the summary. For example, a summary of a photo collection containing both mountains and beaches should contain images of both scene types.

**Diversity:** Summaries should be as diverse as possible, i.e., summaries should not contain images that are similar or identical to each other. Other words for this concept include diversity or dispersion. In computer vision, this property has been referred to as *orthogonality* [28].

Note that [28] also includes the notion of "likelihood," where summary images should have high similarity to many other images in the collection. We believe, however, that such likelihood is covered by fidelity. I.e., a summary that only has images similar to many in the collection might miss certain outlier, or minority, concepts in the collection, while a summary that has high fidelity should include a representative image for every both majority and minority concept in the collection. Also, the above properties could be made very high without imposing further size or budget constraints. I.e., the goal of a summary is to find a small or within-budget subset having the above properties.

## 2.1 Problem Formulation

We cast the problem of image collection summarization as a subset selection problem: given a collection of images $\mathcal{I} = (I_1, I_2, \cdots, I_{|V|})$ represented by an index set $V$ and given a budget $c$, we aim to find a subset $S \subseteq V, |S| \leq c$, which best summarizes the collection. Though alternative approaches are possible, we aim to solve this problem by learning a scoring function $F \colon 2^V \to \mathbb{R}_+$, such that high quality summaries are mapped to high scores and low quality summaries to low scores. Then, image collection summarization can be performed by computing:

$$S^* \in \mathrm{argmax}_{S \subseteq V, |S| \leq c} F(S). \tag{1}$$

For arbitrary set functions, computing $S^*$ is intractable, but for monotone submodular functions we rely on the classic result [25] that the greedy algorithm offers a constant-factor mathematical quality guarantee. Computational tractability notwithstanding, submodular functions are natural for measuring fidelity and diversity [19] as we argue in Section 4.

## 2.2 Evaluation Criteria: V-ROUGE

Before describing practical submodular functions for mixture components, we discuss a crucial element for both summarization evaluation and for the automated learning of mixtures: an objective evaluation criterion for judging the quality of summaries. Our criterion is constructed similar to the popular ROUGE score used in multi-document summarization [17] and that correlates well with human perception. For document summarization, ROUGE (which in fact, is submodular [19, 20]) is defined as:

$$r_{\mathcal{S}}(A) = \frac{\sum_{w \in \mathcal{W}} \sum_{S \in \mathcal{S}} \min\left(c_w(A), c_w(S)\right)}{\sum_{w \in \mathcal{W}} \sum_{S \in \mathcal{S}} c_w(S)} \; (\triangleq r(A) \text{ when } \mathcal{S} \text{ is clear from the context}), \tag{2}$$

where $\mathcal{S}$ is a set of human-generated reference summaries, $\mathcal{W}$ is a set of features ($n$-grams), and where $c_w(A)$ is the occurrence-count of $w$ in summary $A$. We may extend $r(\cdot)$ to handle images by letting $\mathcal{W}$ be a set of visual words, $\mathcal{S}$ a set of reference summaries, and $c_w(A)$ be the occurrence-counts of visual word $w$ in summary $A$. Visual words can for example be computed from SIFT-descriptors [21] as common in the popular bag-of-words framework in computer vision [31]. We call this V-ROUGE (visual ROUGE). In our experiments, we use visual words extracted from color histograms, from super-pixels, and also from OverFeat [27], a deep convolutional network — details are given in Section 5.

## 3 Learning Framework

We construct our submodular scoring functions $F_w(\cdot)$ as convex combinations of non-negative submodular functions $f_1, f_2, \ldots, f_m$, i.e. $F_w(S) = \sum_{i=1}^m w_i f_i(S)$, where $w = (w_1, \ldots, w_m)$, $w_i \geq 0, \sum_i w_i = 1$. The functions $f_i$ are submodular *components* and assumed to be normalized: i.e., $f_i(\emptyset) = 0$, and $f_i(V) = 1$ for polymatroid functions and $\max_{A \subseteq V} f_i(A) \leq 1$ for non-monotone functions. This ensures that the components are *compatible* with each other. Obviously, the merit of the scoring function $F_w(\cdot)$ depends on the selection of the components. In Section 4, we provide a large number of natural component choices, mixtures over which span a large diversity of submodular functions. Many of these component functions have appeared individually in past work and are unified into a single framework in our approach.

**Large-margin Structured Prediction:** We optimize the weights $w$ of the scoring function $F_w(\cdot)$ in a large-margin structured prediction framework, i.e. the weights are optimized such that human summaries $\mathcal{S}$ are separated from competitor summaries by a loss-dependent margin:

$$F_w(S) \geq F_w(S') + \ell(S'), \quad \forall S \in \mathcal{S}, S' \in \mathcal{Y} \setminus \mathcal{S}, \tag{3}$$

where $\ell(\cdot)$ is the considered loss function, and where $\mathcal{Y}$ is a structured output space (for example $\mathcal{Y}$ is the set of summaries that satisfy a certain budget $c$, i.e. $\mathcal{Y} = \{S' \subseteq V : |S'| \leq c\}$). We assume

the loss to be normalized, $0 \leq \ell(S') \leq 1, \forall S' \subseteq V$, to ensure mixture and loss are calibrated. Equation (3) can be stated as $F_w(S) \geq \max_{S' \in \mathcal{Y}} [F_w(S') + \ell(S')], \forall S \in \mathcal{S}$ which is called *loss-augmented inference*. We introduce slack variables and minimize the regularized sum of slacks [20]:

$$\min_{w \geq 0, \|w\|_1 = 1} \quad \sum_{S \in \mathcal{S}} \left[ \max_{S' \in \mathcal{Y}} [F_w(S') + \ell(S')] - F_w(S) \right] + \frac{\lambda}{2} \|w\|_2^2, \tag{4}$$

where the non-negative orthant constraint, $w \geq 0$, ensures that the final mixture is submodular. Note a 2-norm regularizer is used on top of a 1-norm constraint $\|w\|_1 = 1$ which we interpret as a prior to encourage higher entropy, and thus more diverse mixture, distributions. Tractability depends on the choice of the loss function. An obvious choice is $\ell(S) = 1 - r(S)$, which yields a non-submodular optimization problem suitable for optimization methods such as [10] (and which we try in Section 7). We also consider other loss functions that retain submodularity in loss augmented inference. For now, assume that $\tilde{S} = \max_{S' \in \mathcal{Y}} [F_w(S') + \ell(S')]$ can be estimated efficiently. The objective in (4) can then be minimized using standard stochastic gradient descent methods, where the gradient for sample $S$ with respect to weight $w_i$ is given as

$$\frac{\partial}{\partial w_i} \left( F_w(\tilde{S}) + \ell(\tilde{S}) - F_w(S) + \frac{\lambda}{2} \|w\|_2^2 \right) = f_i(\tilde{S}) - f_i(S) + \lambda w_i. \tag{5}$$

**Loss Functions:** A natural loss function is $\ell_{1-R}(S) = 1 - r(S)$ where $r(S) = $ V-ROUGE$(S)$. Because $r(S)$ is submodular, $1 - r(S)$ is supermodular and hence maximizing $F_w(S') + \ell(S')$ requires difference-of-submodular set function maximization [24] which is NP-hard [10]. We also consider two alternative loss functions [20], *complement V-ROUGE* and *surrogate V-ROUGE*. Complement V-ROUGE sets $\ell_c(S) = r(V \setminus S)$ and is still submodular but it is non-monotone. $\ell_c(\cdot)$ does have the necessary characteristics of a proper loss: summaries $S_+$ with large V-ROUGE score are mapped to small values and summaries $S_-$ with small V-ROUGE score are mapped to large values. In particular, submodularity means $r(S) + r(V \setminus S) \geq r(V) + r(\emptyset) = r(V)$ or $r(V \setminus S) \geq r(V) - r(S) = 1 - r(S)$, so complement rouge is a submodular upper bound of the ideal supermodular loss. We define surrogate V-ROUGE as $\ell_{\text{surr}}(A) = \frac{1}{Z} \sum_{S \in \mathcal{S}} \sum_{w \in \mathcal{W}_S^c} c_w(A)$, where $\mathcal{W}_S^c$ is the set of visual words that do not appear in reference summary $S$ and $Z$ is a normalization constant. Here, a summary has a high loss if it contains many visual words that do not occur in reference summaries and a low loss if it mainly contains visual words that occur in the reference summaries. Surrogate V-ROUGE is not only monotone submodular, it is modular.

**Loss augmented Inference:** Depending on the loss function, different algorithms for performing loss augmented inference, i.e. computation of the maximum in (4), must be used. When using the surrogate loss $l_{\text{surr}}(\cdot)$, the mixture function together with the loss, i.e. $f_L(S) = F_w(S) + \ell(S)$, is submodular and monotone. Hence, the greedy algorithm [25] can be used for maximization. This algorithm is extremely simple to implement, and starting at $S^0 = \emptyset$, iteratively chooses an element $j \notin S^t$ that maximizes $f_L(S^t \cup j)$, until the budget constraint is violated. While the complexity of this simple procedure is $O(n^2)$ function evaluations, it can be significantly accelerated, thanks again to submodularity [23], which in practice we find is almost linear time. When using complement rouge $\ell_c(\cdot)$ as the loss, $f_L(S)$ is still submodular but no longer monotone, so we utilize the randomized greedy algorithm [2] (which is essentially a randomized variant of the greedy algorithm above, and has approximation guarantees). Finally, when using loss 1-V-ROUGE, $F_w(S) + \ell(S)$ is neither submodular nor monotone and approximate maximization is intractable. However, we resort to well motivated and scalable heuristics, such as the submodular-supermodular procedures that have shown good performance in various applications [24, 10].

**Runtime Inference:** Having learnt the weights for the mixture components, the resulting function $F_w(S) = \sum_{i=1}^m w_i f_i(S)$ is monotone submodular, which can be optimized by the accelerated greedy algorithm [23]. Thanks to submodularity, we can obtain near optimal solutions very efficiently.

## 4  Submodular Component Functions

In this section, we consider candidate submodular component functions to use in $F_w(\cdot)$. We consider first functions capturing more of the notion of fidelity, and then next diversity, although the distinction is not entirely crystal clear in these functions as some have aspects of both. Many of the components are graph-based. We define a weighted graph $G(V, E, s)$, with $V$ representing a the full set of images and $E$ is every pair of elements in $V$. Each edge $(i, j) \in E$ has weight $s_{i,j}$ computed from the visual features as described in Section 7. The weight $s_{i,j}$ is a similarity score between images $i$ and $j$.

## 4.1 Fidelity-like Functions

A function representing the fidelity of a subset to the whole is one that gets a large value when the subset faithfully represents that whole. An intuitively reasonable property for such a function is one that scores a summary highly if it is the case that the summary, as a whole, is similar to a large majority of items in the set $V$. In this case, if a given summary $A$ has a fidelity of $f(A)$, then any superset $B \supset A$ should, if anything, have higher fidelity, and thus it seems natural to use only monotone non-decreasing functions as fidelity functions. Submodularity is also a natural property since as more and more properties of an image collection are covered by a summary, the less chance any given image not part of the summary would have in offering additional coverage — in other words, submodularity is a natural model for measuring the inherent redundancy in any summary. Given this, we briefly describe some possible choices for coverage functions:

**Facility Location.** Given a summary $S \subseteq V$, we can quantify coverage of the whole image collection $V$ by the similarity between $i \in V$ and its closest image $j \in S$. Summing these similarities yields the facility location function $f_{\text{fac.loc.}}(S) = \sum_{i \in V} \max_{j \in S} s_{i,j}$. The facility location function has been used for scene summarization in [28] and as one of the components in [20].

**Sum Coverage.** Here, we compute the average similarity in $S$ rather than the similarity of the best element in $S$ only. From the graph perspective $(G)$ we sum over the weights of edges with at least one vertex in $S$. Thus, $f_{\text{sum cov.}}(S) = \sum_{i \in V} \sum_{j \in S} s_{i,j}$.

**Thresholded sum/truncated graph cut** This function has been used in document summarization [20] and is similar to the sum coverage function except that instead of summing over all elements in $S$, we threshold the inner sum. Define $\sigma_i(S) = \sum_{j \in S} s_{i,j}$, i.e. informally, $\sigma_i(S)$ conveys how much of image $i$ is covered by $S$. In order to keep an element $i$ from being overly covered by $S$ as the cause of the objective getting large, we define $f_{\text{thresh.sum}}(S) = \sum_{i \in V} \min(\sigma_i(S), \alpha \, \sigma_i(V))$, which is both monotone and submodular [20]. Under budget constraints, this function avoids summaries that over-cover any images.

**Feature functions.** Consider a bag-of-words image model where for $i \in V$, $b_i = (b_{i,w})_{w \in \mathcal{W}}$ is a bag-of-words representation of image $i$ indexed by the set of visual words $\mathcal{W}$ (cf. Section 5). We can then define a feature coverage function [14], defined using the visual words, as follows: $f_{\text{feat.cov.}}(S) = \sum_{w \in \mathcal{W}} g\left(\sum_{i \in \mathcal{I}} b_{i,w}\right)$, where $g(\cdot)$ is a monotone non-decreasing concave function. This class is both monotone and submodular, and an added benefit of scalability, since it does not require computation of a $O(n^2)$ similarity matrix like the graph-based functions proposed above.

## 4.2 Diversity

Diversity is another trait of a good summary, and there are a number of ways to quantify it. In this case, while submodularity is still quite natural, monotonicity sometimes is not.

**Penalty based diversity/dispersion** Given a set $S$, we penalize similarity within $S$ by summing over all pairs as follows: $f_{\text{dissim.}}(S) = -\sum_{i \in S} \sum_{j \in S, j > i} s_{i,j}$ [28] (a variant, also submodular, takes the form $-\sum_{i,j \in S} s_{i,j}$ [19]). These functions are submodular, and monotone decreasing, so when added to other functions can yield non-monotone submodular functions. Such functions have occurred before in document summarization [19], as a dispersion function [1], and even for scene summarization [28] (in this last case, the submodularity property was not explicitly mentioned).

**Diversity reward based on clusters.** As in [20], we define a cluster based function rewarding diversity. Given clusters $P_1, P_2, \cdots, P_k$ obtained by some clustering algorithm. We define diversity reward functions $f_{\text{div.reward}}(S) = \sum_{j=1}^{k} g(S \cap P_j)$, where $g(\cdot)$ is a monotone submodular function so that $f_{\text{div.reward}}(\cdot)$ is also monotone and submodular. Given a budget, $f_{\text{div.reward}}(S)$ is maximized by selecting $S$ as diverse, over different clusters, as possible because of diminishing credit when repeatedly choosing an item in a cluster.

## 5 Visual Words for Evaluation

V-ROUGE (see Section 2.2) depends on a visual "bag-of-words" vocabulary, and to construct a visual vocabulary, multitude choices exists. Common choices include SIFT descriptors [21], color descriptors [34], raw image patches [7], etc. For encoding, vector quantization (histogram encoding) [4], sparse coding [35], kernel codebook encoding [4], etc. can all be used. For the construction of our

V-ROUGE metric, we computed three lexical types and used their union as our visual vocabulary. The different types are intended to capture information about the images at different scales of abstraction.

**Color histogram.** The goal here is to capture overall image information via color information. We follow the method proposed in [34]: Firstly, we extract the most frequent colors in RGB color space from the images in an image collection using $10 \times 10$ pixel patches. Secondly, these frequent colors are clustered by k-means into 128 clusters, resulting in 128 cluster centers. Finally, we quantize the most frequent colors in every $10 \times 10$ pixel image patch using nearest neighbor vector quantization. For every image, the resulting bag-of-colors is normalized to unit $\ell_1$-norm.

**Super pixels.** Here, we wish to capture information about small objects or image regions that are identified by segmentation. Images are first segmented using the quick shift algorithm implemented in VLFeat [33]. For every segment, dense SIFT descriptors are computed and clustered into 200 clusters. Then, a patch-wise intermediate bag of words $b_{\text{patch}}$ is computed by vector quantization and the RGB color histogram of the corresponding patch $c_{\text{patch}}$ is appended to that set of words. This results in intermediate features $\phi_{\text{patch}} = [b_{\text{patch}}, c_{\text{patch}}]$. These intermediate features are again clustered into 200 clusters. Finally, the intermediate features are vector-quantized according to their $\ell_1$-distance. This final bag-of-words representation is normalized to unit $\ell_1$-norm.

**Deep convolutional neural network.** Our deep neural network based words are meant to capture high-level information from the images. We use OverFeat [27], i.e. an image recognizer and feature extractor based on a convolutional neural network for extracting medium to high level image features. A sliding window is moved across an input picture such that every image is divided into $10 \times 10$ blocks (using a $50\%$ overlap) and the pixels within the window are presented to OverFeat as input. The activations on layer 17 are taken as intermediate features $\phi_k$ and clustered by k-means into 300 clusters. Then, each $\phi_k$ is encoded by kernel codebook encoding [4]. For every image, the resulting bag-of-words representation is normalized to the unit $\ell_1$-norm.

## 6  Data Collection

**Dataset.** One major contribution of our paper is our new data set which we plan soon to publicly release. Our data set consists of 14 image collections, each comprising 100 images. The image collections are typical real world personal image collections as they, for the most part, were taken during holiday trips. For each collection, human-generated summaries were collected using Amazon mechanical Turk. Workers were asked to select a subset of 10 images from an image collection such that it summarizes the collection in the best possible way.[1] In contrast to previous work on movie summarization [13], Turkers were not tested for their ability to produce high quality summaries. Every Turker was rewarded 10 US cents for every summary.

**Pruning of poor human-generated summaries.** The summaries collected using Amazon mechanical Turk differ drastically in quality. For example, some of the collected summaries have low quality because they do not represent an image collection properly, e.g. they consist only of pictures of the same people but no pictures showing, say, architecture. Even though we went through several distinct iterations of summary collection via Amazon Turk, improving the quality of our instructions each time, it was impossible to ensure that all individuals produced meaningful summaries. Such low quality summaries can drastically degrade performance of the learning algorithm. We thus developed a strategy to automatically prune away bad summaries, where "bad" is defined as the worst V-ROUGE score relative to a current set of human summaries. The strategy is depicted in Algorithm 1. Each pruning step removes the worst human summary, and then creates a new instance of V-ROUGE using the updated pruned summaries. Pruning proceeds as long as a significant fraction (greater than a desired "$p$-value") of null-hypothesis summarizes (generated uniformly at random) scores better than the worst human summary. We chose a significant value of $p = 0.10$.

## 7  Experiments

To validate our approach, we learned mixtures of submodular functions with 594 component functions using the data set described in Section 6. In this data set, all human generated reference summaries are size 10, and we evaluated performance of our learnt mixtures also by producing size 10 summaries. The component functions were the monotone submodular functions described in

**Algorithm 1** Algorithm for pruning poor human-generated summaries.

---

**Require:** Confidence level $p$, human summaries $\mathcal{S}$, number of random summaries $N$

    Sample $N$ uniformly at random size-10 image sets, to be used as summaries $\mathcal{R} = (R_1, \ldots, R_N)$
    Instantiate V-ROUGE-score $r_{\mathcal{S}}(\cdot)$ instantiated with summaries $\mathcal{S}$
    $o \leftarrow \frac{1}{|\mathcal{R}|} \sum_{R \in \mathcal{R}} \mathbf{1}_{\{r_{\mathcal{S}}(R) > \min_{S \in \mathcal{S}} r_{\mathcal{S}}(S)\}}$ // fraction of random summaries better than worst human
    **while** $o > p$ **do**
        $\mathcal{S} \leftarrow \mathcal{S} \setminus (\operatorname{argmin}_{S \in \mathcal{S}} r_{\mathcal{S}}(S))$
        Re-instantiate V-ROUGE score $r_{\mathcal{S}}(\cdot)$ using updated pruned human summaries $\mathcal{S}$.
        Recompute overlap $o$ as above, but with updated V-ROUGE score.
    **end while**
    **return** Pruned human summaries $\mathcal{S}$

---

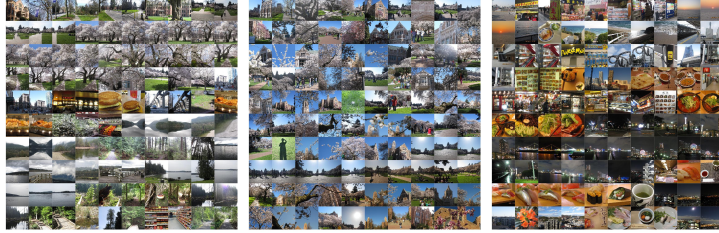

Figure 1: Three example $10 \times 10$ image collections from our new data set.

Section 4 using features described in Section 5. For weight optimization, we used AdaGrad [6], an adaptive subgradient method allowing for informative gradient-based learning. We do 20 passes through the samples in the collection.

We considered two types of experiments: 1) cheating experiments to verify that our proposed mixture components can effectively learn good scoring functions; and 2) a 14-fold cross-validation experiment to test our approach in real-world scenarios. In the cheating experiments, training and testing is performed on the same image collection, and this is repeated 14 times. By contrast, for our 14-fold cross-validation experiments, training is performed on 13 out of 14 image collections and testing is performed on the held out summary, again repeating this 14 times. In both experiment types, since our learnt functions are always monotone submodular, we compute summaries $S^*$ of size 10 that approximately maximize the scoring functions using the greedy algorithm. For these summaries, we compute the V-ROUGE score $r(S^*)$. For easy score interpretation, we normalize it according to $\mathrm{sc}(S^*) = (r(S^*) - \overline{R})/(\overline{H} - \overline{R})$, where $\overline{R}$ is the average V-ROUGE score of random summaries (computed from 1000 summaries) and where $\overline{H}$ is the average V-ROUGE score of the collected final pruned human summaries. The result $\mathrm{sc}(S^*)$ is smaller than zero if $S^*$ scores worse than the average random summary and larger than one if it scores better than the average human summary.

The best cheating results are shown as *Cheat* in Table 1, learnt using 1-V-ROUGE as a loss. The results in column *Min* are computed by constrainedly minimizing V-ROUGE via the methods of [11], and the results in column *Max* are computed by maximizing V-ROUGE using the greedy algorithm. Therefore, the *Max* column is an approximate upper bound on our achievable performance. Clearly, we are able to learn good scoring functions, as on average we significantly exceed average human performance, i.e., we achieve an average score of 1.42 while the average human score is 1.00.

Results for cross-validation experiments are presented in Table 1. In the columns *Our Methods* we present the performance of our mixtures learnt using the proposed loss functions described in Section 3. We also present a set of baseline comparisons, using similarity scores computed via a histogram intersection [32] method over the visual words used in the construction of V-ROUGE. We present baseline results for the following schemes:

    FL   the facility location objective $f_{\text{fac.loc.}}(S)$ alone;

   $\text{FL}_{\text{pen}}$   the facility location objective mixed with a $\lambda$-weighted penalty, i.e. $f_{\text{fac.loc.}}(S) + \lambda f_{\text{dissim.}}(S)$;

  MMR   Maximal marginal relevance [3], using $\lambda$ to tradeoff between relevance and diversity;

 $\text{GC}_{\text{pen}}$   Graphcut mixed with a $\lambda$-weighted penalty, similar to $\text{FL}_{\text{pen}}$ but where graphcut is used in place of facility location;

    kM   K-Medoids clustering [9, Algorithm 14.2]. Initial cluster centers were selected uniformly at random. As a dissimilarity score between images $i$ and $j$, we used $1 - s_{i,j}$. Clustering was run 20 times, and we used the cluster centers of the best clustering as the summary.

In each of the above cases where a $\lambda$ weight is used, we take for each image collection the $\lambda \in \{0, 0.1, 0.2, \ldots, 0.9, 1.0\}$ that produced a submodular function that when maximized produced the best average V-ROUGE score on the 13 training image sets. This approach, therefore, selects the best baseline possible when performing a grid-search on the training sets. Note that both $\lambda$-dependent functions, i.e. $FL_{pen}$ and $GC_{pen}$, are non-monotone submodular. Therefore, we used the randomized greedy algorithm [2] for maximization which has a mathematical guarantee (we ran the algorithm 10 times and used the best result).

Table 1 shows that using 1-V-ROUGE as a loss significantly outperforms the other methods. Furthermore, the performance is on average better than human performance, i.e. we achieve an average score of 1.13 while the average human score is 1.00. This indicates that we can efficiently learn scoring functions suitable for image collection summarization. For the other two losses, i.e. surrogate and complement V-ROUGE, performance is significantly worse. Thus, in this case it seems advantageous to use the proper (supermodular) loss and heuristic optimization (the submodular-supermodular procedure [24, 10]) for loss-augmented inference during training, compared to using an approximate (submodular or modular) loss in combination with an optimization algorithm for loss-augmented inference with strong guarantees. This could, however, perhaps be circumvented by constructing a more accurate strictly submodular surrogate loss but we leave this to future work.

Table 1: Cross-Validation Experiments (see text for details). Average human performance is 1.00, average random performance is 0.00. For each image collection, the best result achieved by any of *Our Methods* and by any of the *Baseline Methods* is highlighted in bold.

| No. | Limits | | | Our Methods | | | Baseline Methods | | | | |
|---|---|---|---|---|---|---|---|---|---|---|---|
| | Min | Max | Cheat | $\ell_{1-R}$ | $\ell_c$ | $\ell_{surr}$ | FL | $FL_{pen}$ | MMR | $GC_{pen}$ | kM |
| 1 | -2.55 | 2.78 | 1.71 | **1.51** | 0.87 | -0.36 | **1.45** | 0.82 | -0.51 | 1.06 | 1.23 |
| 2 | -2.06 | 2.22 | 1.38 | **1.27** | 1.26 | 0.44 | 0.18 | 0.58 | 0.65 | 0.21 | **0.89** |
| 3 | -2.07 | 2.24 | 1.64 | **1.46** | 0.95 | 0.23 | 0.47 | **0.94** | 0.85 | -0.53 | 0.52 |
| 4 | -3.20 | 2.04 | 1.42 | **1.04** | 0.81 | -0.18 | 0.71 | 1.01 | 0.51 | -0.02 | **1.32** |
| 5 | -1.65 | 1.92 | 1.60 | **1.11** | 1.06 | 0.58 | **0.96** | 0.93 | 0.95 | -1.28 | 0.70 |
| 6 | -2.83 | 2.40 | 1.81 | **1.47** | 0.65 | 0.27 | **1.26** | 1.16 | -0.08 | 0.20 | 1.05 |
| 7 | -2.44 | 2.07 | 1.07 | **1.07** | 0.96 | 0.15 | 0.93 | 0.70 | -0.33 | -0.84 | **0.97** |
| 8 | -1.66 | 2.04 | 1.45 | **1.13** | 0.96 | 0.07 | 0.62 | 0.38 | 0.57 | -1.27 | **0.91** |
| 9 | -2.32 | 2.59 | 1.73 | **1.21** | 1.13 | 0.51 | 0.81 | **0.94** | 0.09 | -0.59 | 0.38 |
| 10 | -1.46 | 2.34 | 1.39 | **1.06** | 0.78 | 0.14 | **1.58** | 0.99 | -0.26 | 0.07 | 0.73 |
| 11 | -1.55 | 1.85 | 1.22 | **0.95** | 0.92 | -0.08 | 0.43 | **0.56** | -0.29 | 0.05 | 0.26 |
| 12 | -1.74 | 2.39 | 1.57 | **1.11** | 0.58 | 0.12 | **0.78** | 0.54 | 0.02 | -0.01 | 0.63 |
| 13 | -0.94 | 1.72 | 0.77 | 0.32 | **0.53** | 0.14 | 0.02 | -0.06 | **0.52** | -0.04 | 0.02 |
| 14 | -1.46 | 1.75 | 1.07 | **1.08** | 0.97 | 0.77 | 0.23 | 0.14 | 0.22 | -0.80 | **0.29** |
| Avg. | -2.00 | 2.17 | 1.42 | **1.13** | 0.89 | 0.20 | **0.75** | 0.69 | 0.21 | -0.27 | 0.71 |

## 8 Conclusions and Future Work

We have considered the task of automated summarization of image collections. A new data set together with many human generated ground truth summaries was presented and a novel automated evaluation metric called V-ROUGE was introduced. Based on large-margin structured prediction, and either submodular or non-submodular optimization, we proposed a method for learning scoring functions for image collection summarization and demonstrated its empirical effectiveness. In future work, we would like to scale our methods to much larger image collections. A key step in this direction is to consider low complexity and highly scalable classes of submodular functions. Another challenge for larger image collections is how to collect ground truth, as it would be difficult for a human to summarize a collection of, say, 10,000 images.

**Acknowledgments:** This material is based upon work supported by the National Science Foundation under Grant No. (IIS-1162606), the Austrian Science Fund under Grant No. (P25244-N15), a Google and a Microsoft award, and by the Intel Science and Technology Center for Pervasive Computing. Rishabh Iyer is also supported by a Microsoft Research Fellowship award.

## Footnotes

[1] We did not provide explicit instructions on precisely how to summarize an image collection and instead only asked that they choose a representative subset. We relied on their high-level *intuitive* understanding that the gestalt of the image collection should be preserved in the summary.

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
