[Reviews · NeurIPS 2014]

Submitted by Assigned_Reviewer_9

Paper describes a method to produce summaries (i.e. distinctive subsets) of image collections using a learned mixture of sub modular functions. The summaries are evaluated with a variant of the ROUGE evaluation criterion, and outperform a variety of reasonable baselines (including human summaries, which is likely evidence that the problem is hard to formulate or that the criterion isn't perfect). Authors will release a dataset of 14 image collections containing 100 images each, and Turk summaries of these collections on acceptance.

Minor point: l180, "inference in this case is maximizing an upper bound of the loss" - by reference to eqn 3, 4, I can tell what you're trying to say, but this wording isn't that helpful and troubled me for a bit.

Generally, paper strikes me as interesting, but not very. Some of my concern is because image summarization is so nebulous a problem. It's really hard to be sure about what engineering parameters are important: should one be aiming to summarize large collections or small ones? how accurate should summaries be to be useful? is there much value in having "good" summaries? what do users/the industry want? It's also hard to be sure how to evaluate. Authors have tried to do the right
thing, and deserve credit for this, but is ROUGE/V-ROUGE effective at capturing what is really required? The learning components of this paper are a routine application of known methods; what is novel is the feature constructions and the application of the features and methods to the problem. One could reasonably regard this paper as a more natural submission to an information retrieval conference, e.g. SIGIR.
Summary: Paper has merit, and is of interest, but is not a slam-dunk accept. The learning components of this paper are a routine application of known methods; what is novel is the feature constructions and the application of the features and methods to the problem. One could reasonably regard this paper as a more natural submission to an information retrieval conference, e.g. SIGIR.

Submitted by Assigned_Reviewer_22

This paper uses a mixture of submodular functions to address the image summarization problem. A metric called V-ROUGE is proposed to score and evaluate different summarization. A new dataset is collected to support future research.

Although submodularity has been observed in the previous works, this paper makes a solid step in using submodular functions for image summarization. What's more, a evaluation metric is very useful for future research in this field and the proposed one may be a good direction.

There are still some questions to make the paper complete. Some further study is necessary to understand V-ROGUE. It is not clear how much V-ROGUE resembles human perception. It is more convincing if experiments can be conducted in lab or on Amazon Mechanical Turk to get some quantitive results about correlation between V-ROGUE and human perception. Also, although in natural language the concept of "word" is intuitive, it is not clear what's a good visual "word" for ROGUE. Taking a look at scores from different visual words should be a good start.

A series of functions are used in the mixture. It would give us more insights into the problem if the coefficients of different functions were studied. As we have not yet fully understood V-ROGUE, a evaluation based on human perception is necessary to compare different methods.

The experiments were done on a new dataset with 14 groups. More experiments are necessary to show the amount of data is significant enough for training and testing.

All in all, the paper tried to make a step in the right direction, but more experiments are necessary to make it more conclusive.
Summary: The paper makes valuable contribution to image summarization, but there are still some questions to answer for the proposed method and metric.

Submitted by Assigned_Reviewer_29

This paper formulates the task of summarizing personal photo collections as submodular optimization, and proposes a max-margin formulation with multiple choices for loss functions to learn weights to these. A dataset of image collections with several human-generated summaries is presented, together with a scoring method, using which the proposed approach is evaluated. Experiments show that on this dataset, the method has a very good performance, even outperforming human labelers on average.

Quality

This paper is a through investigation of the possible application of submodularity for image summarization. Extensive experiments were conducted to compare different optimization methods (suitable for different choices of loss functions). The authors also provide an ample amount of supplementary material, giving even more details.

Clarity

The paper is well organized, the explanations are clear and also well-written in terms of English usage.

A few remarks:
- on line 144, does the term "compatible" have a special meaning, justifying the italics? (also, similar was the term "calibrate" on line 155.)
- the relation between the components in section 6 of the supplementary material and the summary in section 5 of the paper could be clarified (or at least it could be mentioned that further information can be found in the supplementary material).

Originality

A major contribution of the paper is posing image set summarization as a submodular optimization problem; to the knowledge of this reviewer, this is a novel view of the problem. Together with a new dataset and the adaptation of ROUGE to a significantly different application domain, this paper has several novel contributions to the state of the art.

Significance

According to this reviewer, this work fits well in the topics of interest for NIPS, to which it makes a significant contribution.
Summary: The paper presents an elegant formulation of the problem of image collection summarization along with a new dataset and an evaluation metric. According to this reviewer, the quality, presentation and contributions of the paper make it a good choice for publication at NIPS.
Author Feedback
Author rebuttal: We thank the reviewers for their valuable comments! Before addressing the comments of the individual reviewers, we want to make some general remarks:

A concern raised by the reviewers is about the proposed evaluation metric, i.e. does V-ROUGE correlate well with human perception. We address this issue in the supplementary material of our paper (see Section 5). We do not do a re-evaluation to study correlation, but demonstrate how the quantification provided by V-ROUGE correlates with what we as humans would expect from a summary. As future work, we would like to do an extensive re-evaluation of V-ROUGE to demonstrate the correlation quantitatively. In fact, we have already set up (before we received the NIPS reviews) a website for collecting human ratings for image collection summaries. We are collecting these ratings, and will quantitatively evaluate the correlation of V-ROUGE to human perception.

Note, as mentioned in the paper, V-ROUGE has advantages over criteria proposed in the literature, e.g. bag-of-words reconstruction errors in l2-norm sense [1] -- such criteria are not related to human perception at all. By contrast, V-ROUGE is necessarily related to human perception through the use of human generated reference summaries. Thus the question whether V-ROUGE is a good criterion depends on the used visual words (note, bag-of-words reconstruction errors [1] may not correlate with human perception even for good visual words). Moreover, since our metric is resembles ROUGE, which has extensively been used in document summarization, we feel that this is a good fit for our problem.

Image collection summarization is a challenging problem to solve, and we have made some progress. There might not be a unique best solution, especially because human perception varies. In any case, to compare different algorithms for image collection summarization, some criterion for automatic evaluation is very useful, and we have provided one (the same issues arise in other domains, e.g. speech enhancement, speech synthesis, document summarization, machine translation, etc.).

[1] Yang et al., "Image collection summarization via dictionary learning for sparse representation," CVPR, 2012

# Assigned_Reviewer_22 #

> [...] It is not clear how much V-ROGUE resembles human perception. [...]

Please consider our general remarks.

> Also, although in natural language the concept of "word" is intuitive, it is not clear what's a good visual "word" for ROGUE.

We performed such experiments during the development of V-ROUGE. Initially we did not use visual words extracted from a deep convolutional feature extractor (OverFeat). In this case, we did not achieve performance on par with average humans. This indicates that meaningful higher level information is extracted by OverFeat. We will add results in this regard to the supplement.

> [...] It would give us more insights into the problem if the coefficients of different functions were studied.

We will add information in this regard to the supplement, e.g. histograms of weights for different function classes. Generally, the coverage-functions received a lot of weight.

> As we have not yet fully understood V-ROGUE, a evaluation based on human perception is necessary to compare different methods.

Please consider our general remarks.

> The experiments were done on a new dataset with 14 groups. More experiments are necessary to show the amount of data is significant enough for training and testing. [...]

As explained in the paper (Section 7), we ran 14-fold cross-validation, meaning we train on 13 image sets and test on the 14th, doing that process 14 times. Given that in each case, the performance in the held out 14th image collection improved, this indicates that the information within the data is enough to train and generalize. We believe this is well explained in Section 7 in the paper.

# Assigned_Reviewer_29 #

> on line 144, does the term "compatible" have a special meaning, justifying the italics? [...]

We used italics to indicate that the exact meaning of the terms "compatible" and "calibrate" is not specified. We will clarify in the final camera ready, if given the opportunity.

> the relation between the components in section 6 of the supplementary material and the summary in section 5 of the paper could be clarified [...]

In the final version, if accepted, we will add a reference to the supplement as suggested.

# Assigned_Reviewer_9 #

> Minor point l180:
Thank you for pointing this out. We'll hopefully have the chance to clarify this in the final version of the paper.

> [...] It's really hard to be sure about what engineering parameters are important: should one be aiming to summarize large collections or small ones? how accurate should summaries be to be useful? is there much value in having "good" summaries? [...]

Ideally, we should be able to summarize image collections of any size and have a parameter allowing users to specify the size of the summary to create. The question of how good summaries should be is hard to answer. However, it is clear that we should avoid producing poor summaries. Having good summaries is important: consider for example image search; one expects, as a result, a diverse collection of images relevant to a query, and not to be given the "same result" many times.

> [...], but is ROUGE/V-ROUGE effective at capturing what is really required?

Please refer to the general remarks.

> The learning components of this paper is routine; what is novel is the feature constructions and the application of the features and methods to the problem.

We wish to emphasize that the comparison of different loss functions, i.e. submodular and non-submodular losses, has not been performed as far as we know, and hence we consider this to be a novel component of learning.